# Cancer Cell’s Achilles Heels: Considerations for Design of Anti-Cancer Drug Combinations

**DOI:** 10.3390/ijms252413495

**Published:** 2024-12-17

**Authors:** Valid Gahramanov, Frederick S. Vizeacoumar, Alain Morejon Morales, Keith Bonham, Meena K. Sakharkar, Santosh Kumar, Franco J. Vizeacoumar, Andrew Freywald, Michael Y. Sherman

**Affiliations:** 1Molecular, Cellular, and Developmental Biology, Yale University, New Haven, CT 06511, USA; valid.gahramanov@yale.edu; 2Department of Molecular Biology, Ariel University, Ariel 40700, Israel; shashisantosh2007@gmail.com; 3Department of Pathology and Laboratory Medicine, College of Medicine, University of Saskatchewan, Royal University Hospital, Saskatoon, SK S7N 0W8, Canada; frederick.vizeacoumar@usask.ca (F.S.V.); alm318@mail.usask.ca (A.M.M.); andrew.freywald@usask.ca (A.F.); 4Department of Biochemistry, Microbiology and Immunology, University of Saskatchewan, 107 Wiggins Road, Saskatoon, SK S7N 5E5, Canada; 5Cancer Research, Saskatchewan Cancer Agency and Division of Oncology, University of Saskatchewan, 107 Wiggins Road, Saskatoon, SK S7N 5E5, Canada; bonhamk274@gmail.com (K.B.); franco.vizeacoumar@usask.ca (F.J.V.); 6Drug Discovery and Development Research Group, College of Pharmacy and Nutrition, University of Saskatchewan, 107 Wiggins Road, Saskatoon, SK S7N 5E5, Canada; meena.sakharkar@usask.ca

**Keywords:** shRNA screening, GSEA, drug combination, CMAP, signaling pathway, synergy

## Abstract

Loss of function screens using shRNA (short hairpin RNA) and CRISPR (clustered regularly interspaced short palindromic repeats) are routinely used to identify genes that modulate responses of tumor cells to anti-cancer drugs. Here, by integrating GSEA (Gene Set Enrichment Analysis) and CMAP (Connectivity Map) analyses of multiple published shRNA screens, we identified a core set of pathways that affect responses to multiple drugs with diverse mechanisms of action. This suggests that these pathways represent “weak points” or “Achilles heels”, whose mild disturbance should make cancer cells vulnerable to a variety of treatments. These “weak points” include proteasome, protein synthesis, RNA splicing, RNA synthesis, cell cycle, Akt-mTOR, and tight junction-related pathways. Therefore, inhibitors of these pathways are expected to sensitize cancer cells to a variety of drugs. This hypothesis was tested by analyzing the diversity of drugs that synergize with FDA-approved inhibitors of the proteasome, RNA synthesis, and Akt-mTOR pathways. Indeed, the quantitative evaluation indicates that inhibitors of any of these signaling pathways can synergize with a more diverse set of pharmaceuticals, compared to compounds inhibiting targets distinct from the “weak points” pathways. Our findings described here imply that inhibitors of the “weak points” pathways should be considered as primary candidates in a search for synergistic drug combinations.

## 1. Introduction

Drug cocktails are being routinely used in anti-cancer therapies to improve tumor elimination and reduce the chances of relapse and treatment resistance [1]. The majority of drug combinations that have been used in medical practice were selected in trial-and-error-based searches, mostly testing combinations of existing anti-cancer agents in pre-clinical models and patients [2,3]. More recently, there have been attempts to conduct massive in vitro screens for synergistic drug combinations [4,5,6]. Unfortunately, such screens often have had limited success because of insufficient throughput. In addition, investigators have been approaching the problem in a more rational way by predicting synergistic combinations based on the established mechanisms of action of anticancer drugs. For example, in the case of insufficient effects of *ERBB2* inhibition due to the compensatory overexpression of the EGF receptor (EGFR) in cancer cells, administration of the *ERBB2* inhibitor, Herceptin, is often combined with suppressing EGFR activity [7,8,9,10]. Beyond all these strategies, sophisticated computational analyses of treatment-induced transcriptome modifications and machine learning algorithms have been applied to find novel potent drug combinations [5,6,11,12]. So far, these efforts have not produced a sufficient number of effective drug combinations for most cancer types, especially for advanced and metastatic tumors. Therefore, new research approaches need to be developed.

The published literature reports systematic analyses of 500 RNAi screens and over 300 CRISPR-Cas9 screens conducted to develop a Cancer Dependency Map (Cancer DepMap) [13,14]. This map aims to identify specific cancer vulnerabilities on a large scale, ultimately supporting personalized treatments tailored to patients’ molecular profiles. Lately, shRNA (RNAi) and CRISPR screens within the Cancer Dependency Map have been employed to comprehensively characterize the cellular effects of newly developed and more traditional therapeutic compounds, and to identify genes associated with resistance to their cancer-suppressing action. Such screens have a strong potential to facilitate a search for novel drug combinations [15,16,17].

In an earlier work, we used shRNA screens to identify genes involved in responses to a novel *Hsp70* inhibitor JG-98 [18] and to an antibiotic gentamycin [19]. Although the studies focused on two chemicals that were entirely different both structurally and in their mechanisms of action, surprisingly, a set of common genes was identified in these two screens. Based on this unexpected finding, we suggest that there may be a core set of genes or signaling pathways that constitute “weak points” of the cell, and depletion of such genes or inhibition of such pathways may make cells vulnerable to a variety of drugs and stressors with different mechanisms of action. Knowledge of these “weak points” could be helpful in designing novel drug cocktails. Here, we addressed the “weak points” hypothesis by analyzing multiple datasets from cancer-related shRNA screens and previously tested drug combinations.

## 2. Results

### 2.1. Functional Enrichment Analyses of shRNA Screens

To determine if certain gene sets are commonly identified as hits in diverse shRNA screens, we pooled hits from shRNA screens against eight unrelated drugs from the GEO database (Appendix A) [18,20,21,22].

In the first approach, these data were subjected to the Gene Set Enrichment Analysis (GSEA) to identify pathways that affect sensitivity to the drugs used in the screens. GSEA is particularly useful in this context because it allows us to analyze whether predefined sets of genes—such as those associated with particular biological pathways or functions—are statistically overrepresented among the hits. By focusing on pathways rather than single genes, GSEA can reveal coordinated activity that may be crucial for understanding drug sensitivity and potential resistance mechanisms. To further explore the data, we performed a correlation analysis to examine the relationship between the size of the shRNA libraries and the number of identified hits, where we observed a linear correlation (Appendix A). This finding suggests that larger shRNA libraries tend to yield more hits, which could impact the depth and scope of pathway identification.

### 2.2. Identification of “Weak Points” Pathways

To find common biological processes, inhibitions of which sensitize to various treatments, we compared lists of pathways that show up on different screens. This approach revealed pathways common in at least three, at least four, and at least five screens (Appendix A). Notably, in the analysis of these pathways, one needs to consider that their designation in GSEA is somewhat arbitrary. In other words, the same genes could belong to different pathways defined in GSEA. Indeed, the analysis of gene sets that constitute pathways indicated in Figure 1A revealed that the designation of T-cell signaling, VEGF signaling, mTOR, cytosolic DNA sensing, pathogenic *E. coli* infection, and Gap-junction pathways was due to the presence of similar gene sets found in shRNA screens. Therefore, we considered these pathways together as the Akt-mTOR pathway. Overall, we found several core pathways, which when suppressed increase the sensitivities of cancer cells to drugs in at least three independent screens. This list included ribosome, spliceosome, transcriptomic, proteasome, cell cycle, Akt-mTOR, starch and sucrose metabolism, and tight junction pathways. Among pathways common in at least four screens, we lost the transcription pathway, and pathways common in at least five screens also did not include the mTOR pathway (Figure 1A, Appendix A).

As a complementary approach, we performed a different analysis by evaluating the overlapping genes identified in the shRNA screens. Despite the high diversity of tested drugs in the screens, there were 856 genes present in at least two screens. We enhanced the stringency of our analysis by concentrating only on the overlapping genes that belong to GSEA-designated pathways. This approach allowed us to narrow our search to 222 candidates (Figure 1B).

By analyzing 222 recurrently found genes using the Molecular Signature Database [24], we identified common pathways that appeared to be responsible for the resistance to several drugs. Overall, with the exception of sucrose metabolism pathways, they were identical to common pathways detected via a direct comparison of pathways in Figure 1A. As discussed above, since the designation of certain pathways was due to the presence of the same genes found in shRNA screens, we considered T-cell signaling, VEGF signaling, mTOR, and Gap-junction pathways collectively as the Akt-mTOR pathway. Similarly, we considered DNA replication as part of the cell cycle pathway. Taken together, our two independent strategies pointed towards seven core pathways including ribosome, spliceosome, transcription, proteasome, cell cycle, Akt-mTOR, and tight junction, inhibition of which sensitized cancer cells to various drugs. Overall, these data indicate that the suppression of certain specific pathways in cancer cells may make them more sensitive to drugs with distinct mechanisms of action and, therefore, these pathways may represent “weak points” of cancer cells. We hypothesized that targeting these pathways with drugs may sensitize cells to diverse anti-cancer therapeutics.

### 2.3. Validation of the Hypothesis of “Weak Points” Pathways

Evidence supporting the hypothesis came from our recent works with pooled shRNA screens to identify genes involved in cell responses to an allosteric *Hsp70* inhibitor JG-98 [18] and a B-cell receptor-associated kinase (*BTK*) inhibitor ibrutinib [25]. In both screens, we obtained multiple hits belonging to the “weak points” pathways. Upon further testing of hits, we found potent synergistic effects of inhibitors of proteasome, Akt-mTOR, and transcription with JG-98 [18]. In the case of ibrutinib, we obtained potent synergistic effects of inhibitors of proteasome and Akt-mTOR [25]. Considering the lack of similarities in both the structure of JG-98 and ibrutinib and their mechanisms of action, these findings strongly support the idea that targeting “weak points” pathways may enhance the anti-cancer effects of diverse therapeutics.

To further validate the hypothesis, we analyzed data from the existing literature on the synergistic effects of drugs. The hypothesis that the inhibition of “weak points” pathways can enhance cells’ sensitivity to a variety of anti-cancer drugs makes a clear prediction—drugs that synergize with inhibitors of the “weak points” pathways should be more diverse compared to sets of drugs that synergize with therapeutics that affect targets other than the “weak points”. Accordingly, we collected published data on the drugs that synergize with “weak point” pathway inhibitors (test therapeutics) and with therapeutics that target other pathways (control therapeutics), and we quantified the diversity of these drug sets. Certain possible biases should be considered in such a metadata analysis, including (a) investigator biases related to preexistent knowledge about drug effects in choosing combinations to test, i.e., non-random choice of tested drug combinations, and (b) bias related to the novelty of tested therapeutics, which could result in only a few reported synergistic combinations (Figure 2A). We attempted to overcome both types of biases by choosing for our analysis only therapeutics for which synergistic combinations were established in more than 30 publications. Based on this, we excluded translation and splicing pathways, since their targeting is new in clinical practice and there is no sufficient number of reported synergistic drugs. The tight junction pathway was excluded because there are no known drugs that target this pathway. Therefore, we selected inhibitors of the proteasome (bortezomib), transcription (*CDK9* inhibitor roscovitine), cell cycle (*CDK4,6* inhibitor abemaciclib), and Akt-mTOR (rapamycin) pathways as test therapeutics for the metadata analysis. Drugs that synergize with inhibitors of these pathways are shown in Appendix A. As control therapeutics that inhibit targets other than the weak points, we chose inhibitors of topoisomerase I (irinotecan), EGFR (erlotinib), genotoxic agent (cisplatin), and antimetabolite (5-fluorouracil).

Our evaluation of the synergistic drug diversity was based on the transcriptomics effects of the drugs. Accordingly, we derived the transcriptomic effects of the synergistic drugs listed in Appendix A from the Connectivity Map project (CMAP) [26]. Since in this database, transcriptomics effects of drugs are compared in multiple cancer lines, we standardized the analysis by choosing the effects of all drugs of interest on a single breast cancer cell line, MCF7. Of note, these drugs showed very similar transcriptomic effects in other cancer cell lines. Upon extraction of the transcriptomics information, we performed a multidimensional scaling analysis [27] to assess similarities and dissimilarities of transcriptomic profiles of drugs that synergize with therapeutics targeting the “weak points” and control pathways. We employed the Euclidean distance (see Section 4) in the MDS plot to statistically assess the diversity score of the therapeutics that synergize with “weak points” and control pathways inhibitors. In the MDS analysis, for control therapeutics, we excluded drugs that affect the same pathway as the control therapeutics to reduce the bias towards lower diversity. For test therapeutics, we excluded inhibitors of the other “weak point” pathways since they would artificially increase the diversity score.

As predicted by the hypothesis, the transcriptional diversity of drugs synergized with test therapeutics was higher compared to the diversity of drugs that synergized with control therapeutics, and this difference was highly statistically significant (Figure 2B,C and Appendix A). In other words, the inhibition of the “weak points” pathways indeed sensitized cells to a higher diversity of drugs than the inhibition of other pathways.

These findings support the hypothesis that at least proteasome, transcription, the cell cycle, and Akt-mTOR pathways are Achilles heels of the cell, partial inhibition of which makes cancer cells more sensitive to a wide variety of drugs.

## 3. Discussion

Combination therapies are viewed as commonly used strategies for treating cancer patients [28,29,30,31]. Although combination therapy involving chemotherapeutic agents can be toxic, the overall toxicity is often reduced because different pathways are targeted. Ultimately, this works in a synergistic or additive manner, and, therefore, a lower therapeutic dosage of each individual drug is required. Here, based on multistep bioinformatics analysis of previously published and pooled shRNA screens (shown see Appendix A), we determined that partial depletion of genes belonging to a limited set of pathways enhances sensitivities of cancer cells to various therapeutic compounds. Surprisingly, these drugs were unrelated in their mechanisms of action, suggesting that partial suppression of the identified pathways makes cells more sensitive to general disturbances of their homeostasis. Therefore, we call them the “weak points” of the cell.

Analysis of genes belonging to the “weak points” pathways uncovered a relation to “cancer essentiality genes” [32,33,34]. Indeed, a previous study with a large-scale shRNA screen uncovered 297 genes crucial for the survival and proliferation of 72 cancer cell lines, but not non-malignant cells (so-called cancer essentiality genes) [32]. Indeed, their partial suppression by shRNA-based silencing proved to be selectively lethal for cancer cells [35]. We observed a strong overlap between cancer essentiality pathways and pathways that affect sensitivity to various drugs in our analysis of shRNA screens, including ribosome, splicing, transcriptome, cell cycle, and proteasome.

Why did the shRNA downregulation of these pathways in the publication of cancer essentiality genes [32] lead to the death of cancer cells, while in the screens that we cite here, the shRNA downregulation of these genes did not kill cells but only enhanced the killing by various drugs? Most likely, in the original publication that identified the essentiality genes [32], relatively strong gene silencing was achieved, which was sufficient for the loss of viability of cancer cells. In contrast, the drug-related screens we utilized in this study applied milder gene downregulation, which was not toxic enough to kill the cancer cells directly. However, this milder suppression made the cancer cells more vulnerable to certain compounds, enhancing their sensitivity and enabling synergistic effects with these drugs. Interestingly, two “weak points” pathways, including Akt-mTOR and tight junction, were not found among the cancer essentiality pathways in [32], which highlights a distinct functional role under drug-enhanced conditions rather than an absolute necessity for survival under normal conditions.

The concept of “weak points”, i.e., selected pathways, suppression of which sensitizes cancer cells to a variety of treatments, predicts that inhibitors of these pathways should synergize with a highly diverse set of drugs. To address this question, we compared the diversity of synergistic drugs to inhibitors of the “weak points” pathways and inhibitors of other pathways. In other words, we quantitatively evaluated the diversity of drugs synergistic with these therapies. Originally, it was unclear to us how to perform such an evaluation based on the mechanisms of the drugs’ action, e.g., how to estimate whether the pair of inhibitors of microtubules and EGFR is more or less diverse than the pair of inhibitors of mTOR and topoisomerase I. We approached the diversity based on the transcriptomics effects of the synergistic drugs. An extraction of transcriptomics effects of selected drugs from the CMAP database and MDS plotting indeed allowed us to assign the diversity scores and compare the diversity levels. We realize that functional comparison would probably be more adequate, but as noted above, such a comparison does not allow for scoring. Our data also support the idea that transcriptomic profiles of anti-cancer compounds can be used as an effective tool in the search of effective drug combinations.

More specifically, our work strongly suggests that inhibitors of the identified “weak points” pathways should be given priority as candidates for developing new combination therapies. Indeed, inhibitors of some of the “weak points” pathways are already in clinical practice, e.g., inhibitor of Akt-mTOR capivasertib [36,37,38] or inhibitor of proteasome bortezomib [39,40,41]. We propose that such inhibitors, when used in combination with a novel drug in development, have good chances to increase its efficacy.

## 4. Materials and Methods

### 4.1. Data Resource and Study Design

Data on shRNA screens of drug effects were collected from published [18,20,21,22] and unpublished works represented in the GEO database (see GEO IDs in Appendix A). The shRNA-based drug screens were performed as previously described [42].

Briefly, lentivirus pools were generated from pooled lentiviral plasmid DNA. A total of 4 × 10^7^ cells per replicate were infected with 80 k lentiviral shRNA pools at an MOI of 0.3–0.4. After two days of selection in media containing 2 µg/mL puromycin (Sigma-Aldrich, Waltham, MA, USA) to eliminate uninfected cells, genomic DNA was prepared from shRNA-infected cell populations (Blood Maxi prep kit, Qiagen, Hilden, Germany). Therefore, each hairpin was represented >200 times in the screening populations. Half-hairpin barcodes were prepared from genomic DNA samples using 30 μg of DNA obtained from at least 5 × 10^6^ infected cells so that a 60–70-fold representation was obtained from the starting amount of gDNA. A master mixture for each sample containing 30 μg of template DNA, 2× PCR buffer, 2× enhancer solution, 300 nM each dNTP, 900μM each oligonucleotide primer (PCR_BF 5′-Biotin-AATGGACTATCATATGCTTACCGTAACTTGAA-3′ and PCR_R 5′-TGTGGATGAATACTGCCATTTGTCTCGAGGTC-3′), 50 mM MgSO_4_, 45 units of Platinum Pfx polymerase (Invitrogen, Waltham, MA, USA), and water up to 1200 μL was made and divided into 100 μL aliquots. The amplification reaction was performed by denaturing once at 94 °C for 5 min, followed by (94 °C for 15 s, 55 °C for 15 s, 68 °C for 20 s) ×30, 68 °C for 5 min, and then cooling to 4 °C. The PCR product (178 bps) was run on a 2% agarose gel to make sure that the amplified shRNA sequence did not form a cruciform structure (225 bps). PCR products were immediately purified using the QIAquick PCR purification kit (Qiagen) to avoid the conversion of the linear product to cruciform DNA and immediately digested with XhoI (New England Biolabs, Ipswich, WA, USA) for 2 h at 37 °C to generate a thermo-stable half-hairpin probe. This was then gel-purified, and the remaining salts were cleared using a PCR purification kit (Qiagen) with two elutions of 30 μL of EB buffer (Qiagen). An average yield of 3 to 3.5 μg of each sample was obtained from this procedure. The probe was hybridized onto UT-GMAP 1.0 microarrays (Affymetrix Inc., Santa Clara, CA, USA). Genes, the depletion of which sensitizes to drugs in each screen, are shown in Appendix A.

### 4.2. Gene Set Enrichment Analysis (GSEA) and Connectivity Map (CMAP)

The entire list of sensitizing genes in screens was ranked according to fold change and used as input to Gene Set Enrichment Analysis (GSEA) [23,43]. GSEA was employed against both the hallmark gene-set signature and curated gene sets, which were obtained from the Molecular Signature Database v7.2 (http://www.gsea-msigdb.org/gsea/msigdb/index.jsp, accessed on 1 December 2022). We took into consideration gene sets with a *p*-value < 0.05 and an FDR cutoff of 25%, which is appropriate in GSEA analysis due to the relatively small number of genes in the sets being analyzed. In fact, GSEA recommends using this cutoff in establishing significant pathways [23,44]. In two screens for GSE125021 and *TP53* inhibitors, where the number of hits identified in the original dataset was notably low (below 300), the FDR cutoff was slightly relaxed to 35–40%. This adjustment was implemented to account for the reduced statistical power inherent in datasets with fewer significant hits, thereby allowing for a broader exploration of possible biological signals that may have, otherwise, been missed.

shRNA expression data were collected from the GEO Datasets databases [45]. Transcriptomic data of drug effects were collected from the Connectivity Map databases [26,46]. To make all the transcriptomic profile analyses unbiased, we extracted the transcriptomic effects of each drug after 24 h of treatment in the MCF7 cell line.

### 4.3. Drug Synergy Analysis

PubMed datasets [47] were selected for identifying synergistic drug combinations. We used keywords, e.g., “Proteasome inhibitor” AND “synergy in cancer” to find publications that contain information about effects of drugs synergistic with proteasome inhibitors. Further, we followed the same searching method for the remaining pathway inhibitors—“*CDK9*”, “Cisplatin”, “Irinotecan”, “Rapamycin”, “Cell cycle”.

### 4.4. Multidimensional Scaling

To check the similarity and dissimilarity of transcriptomic effects of drugs, we utilized a multidimensional scaling analysis [27] with edgeR [48] and limma [44] packages in R programming language. The distances of the drugs’ transcriptomic data on the dimensional plot were calculated based on the Euclidean distance algorithm [49,50].

### 4.5. Statistical Analysis

Statistical analysis was performed in R. For the calculation of Euclidian distance, we used the Pythagorean equation—$\sqrt{\sum_{i}((a_{i\;}-b_{i\;}{)}^{2})}$. Observations a and b were measured in many dimensions. Student’s t-test was used to evaluate the statistical significance of the difference between the two groups. A *p*-value less than 0.05 was considered as significant. Statistical differences are representative of the consideration of four observations per group (n = 4).

## 5. Conclusions

Our research highlights specific pathways that serve as “weak points” in cancer cells, the mild disturbance of which is likely to make cells more vulnerable to various stressors and therapeutic drugs. These insights pave the way for identifying optimal combinations of synergistic anti-cancer drugs, offering new strategies to enhance treatment effectiveness.

## Figures and Tables

**Figure 1 ijms-25-13495-f001:**
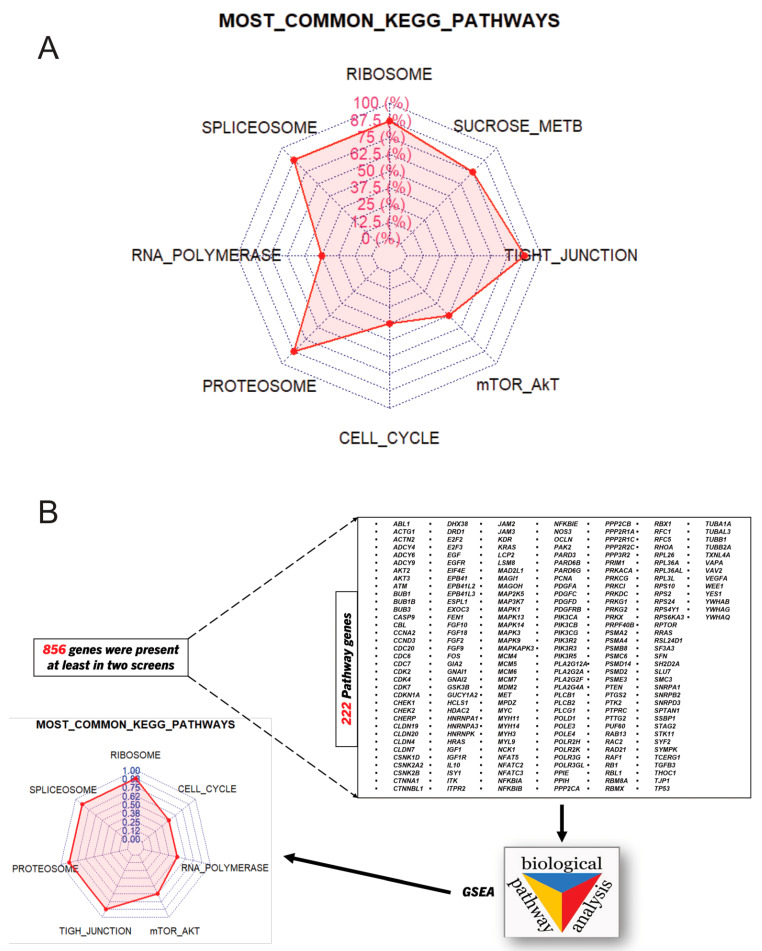
(**A**) Radar plot representing GSEA analysis of sensitizing hits from the shRNA screens. Each blue dotted line represents a screen. The red dots mark the number of screens that contain indicated pathways. FDR values were below 0.25, which is a standard for generating a hypothesis [23]. (**B**) A complementary analysis of shRNA screen hits. Genes common for at least two screens were identified followed by the GSEA analysis. Radar plot representing GSEA analysis.

**Figure 2 ijms-25-13495-f002:**
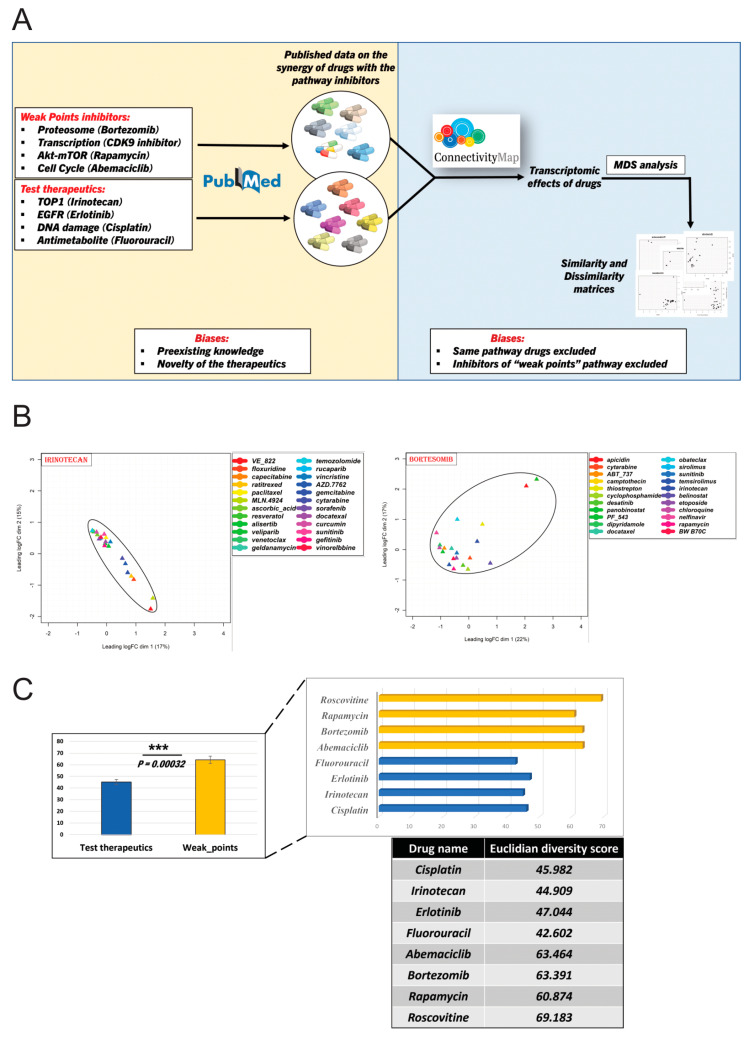
Finding synergistic drugs. (**A**) The graphic represents an overall summary of the analysis of synergistic drugs against “weak points” and regular pathways. (**B**) The MDS plot demonstrates the diversity of drugs that synergize with a “weak point” pathway inhibitor bortezomib and a “control therapeutics” *Top1* inhibitor irinotecan. The area covered by drug effects-representing triangles reflects the diversity of the drugs’ actions. (**C**) Bar plots represent the quantitative diversity of drugs that synergize with inhibitors of “weak points” pathways and control therapeutics. The left plot shows an average of drug diversity calculated based on Euclidian distances between the drugs on the MDS plots. The right plot represents a diversity of drugs synergizing with individual therapeutics. The table shows diversity scores for drugs synergizing with each of the therapeutics. ***—*p*-value < 0.001. Statistical analysis was conducted using Student’s *t*-test by considering 4 drugs per group (n = 4).

## Data Availability

The manuscript contains Appendix A that have been provided separately. Raw and processed data are stored in the GEO database (see Section 4) website. Codes and pipelines will be made available upon request.

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
