# Peer review of "Cancer Cell’s Achilles Heels: Considerations for Design of Anti-Cancer Drug Combinations"

_ijms, 2024, doi:10.3390/ijms252413495_

Round 1

Reviewer 1 Report

Comments and Suggestions for Authors

The search for safe and effective combination therapies for cancer diseases is of great importance. The authors outline the relevance of drugs inhibiting so-called ´´Achilles heel´´ mechanisms and targets in combination with chemotherapy based on previous findings from studies using Hsp70 inhibitor JQ-98. The manuscript is interesting and provides new clues for the design of efficient anticancer combination therapies. However, I have some comments for the authors on the manuscript and I recommend major revision:

Maybe the authors can discuss epigenetic drugs, which are missing in the current manuscript, in terms of transcription inhibition.

It is unclear why a CDK inhibitor was used as transcription inhibitor. Please explain why the CDK9 inhibitor roscovitine was selected as a transcription inhibitor for this study. How far are the effects of roscovitine distinct from the CDK inhibitor abemaciclib (cell cycle inhibitor)?

I’m rather skeptical that the determined ´´Achilles heels´´ are relevant for all cancers. Are there cancer models, which are resistant/insensitive to targeting the described mechanisms?

I suppose the Conclusions should be placed after the Discussion section according to the journal guidelines.

References: Abbreviate the journal names of references No. 4, 6, 23, 30, 38. PNAS and JCO are improper abbreviations. Ref 15 is incomplete.

Author Response

Michael Sherman                
Professor         
December 12, 2024 
International Journal of Molecular Sciences (IJMS) 
Dear Editor: 
Tel. +972-58-781-9472 
sherma1@ariel.ac.il 
Thank you for handling our manuscript “Cancer Cell’s Achilles Heels: Considerations for design 
of anti-cancer drug combinations” #ijms-3377637.  
We are grateful to the reviewers for their comments and corrected the problems accordingly.  
Below is a point-by-point response to the reviewers’ comments. 
In addition, we addressed the structural guidelines and reference issues that were suggested by 
both editor and reviewers. 
We hope that in the present form, the manuscript is suitable for publication. 
We look forward to hearing from you. 
Sincerely yours, 
Michael Sherman, Ph.D. 
Response to Reviewers. 
REVIEWER 1:  
Comment 1: Maybe the authors can discuss epigenetic drugs, which are missing in the current 
manuscript, in terms of transcription inhibition. 
Answer: We would like to point out that the screens referenced (see supplementary files 
Table_S1) in our study include the use of HDAC inhibitors, which are widely recognized as 
epigenetic drugs. HDAC inhibitors impact transcriptional regulation by modifying chromatin 
structure. Their inclusion in the screens highlights their potential as modulators of the 'weak points' 
identified in our study.  
Comment 2: It is unclear why a CDK inhibitor was used as transcription inhibitor. Please explain 
why the CDK9 inhibitor roscovitine was selected as a transcription inhibitor for this study. How 
far are the effects of roscovitine distinct from the CDK inhibitor abemaciclib (cell cycle inhibitor)? 
Answer: Roscovitine binds to the ATP-binding site of CDK9 and inhibits it. CDK9 functions in 
transcription regulation by phosphorylating the CTD domain of RNA polymerase II at sites that 
promote the continuation of elongation following transcription pausing, which is a critical step in 
the process of transcription elongation. Accordingly, CDK9 inhibitors are considered to be 
transcriptional inhibitors. 
On the other hand, abemaciclib is an inhibitor of CDK4 and CDK6, which function in releasing 
G1 checkpoints and entering the S phase. Therefore, inhibition of these enzymes leads to growth 
arrest. Thus, abemaciclib is considered a cell cycle inhibitor. 
Comment 3: I’m rather skeptical that the determined ´´Achilles heels´´ are relevant for all cancers. 
Are there cancer models, which are resistant/insensitive to targeting the described mechanisms? 
Answer: Indeed, there is evidence in the literature that not all cancer models are equally sensitive 
to targeting the “weak points” pathways, such as proteasome or cell cycle. For instance, certain 
cancer subtypes, like some forms of indolent lymphomas, exhibit resistance to proteasome 
inhibitors due to low proteasome activity and dependency (e.g. Kupperman et al., 2010). Similarly, 
resistance to cell cycle inhibitors, such as CDK4/6 inhibitors, has been observed in cancers with 
alternative mechanisms driving proliferation, as highlighted by O`leary et al., 2016 in the context 
of ER+ breast cancer models. These examples underscore the importance of cancer-specific 
context and emphasize that our findings may not universally apply to all cancers.  
Comment 4: I suppose the Conclusions should be placed after the Discussion section according 
to the journal guidelines.  
Answer: Based on journal guidelines sections of the manuscript have been updated and the 
conclusion part was placed after Discussion.  
Comment 5: References: Abbreviate the journal names of references No. 4, 6, 23, 30, 38. PNAS 
and JCO are improper abbreviations. Ref 15 is incomplete. 
Answer: We have addressed the reviewer’s comments by correcting the references and resolving 
the issue of abbreviation in the reference list. The updated references are highlighted in the revised 
manuscript for clarity.

Reviewer 2 Report

Comments and Suggestions for Authors

The manuscript by Gahramanov et al. identified seven core pathways, named the "Achilles heels" of cancer cells. These pathways included the proteasome, protein synthesis, RNA splicing, RNA synthesis, the cell cycle, the Akt-mTOR pathway, and tight junction-related mechanisms. Through the analyses using Gene Set Enrichment Analysis (GSEA) and the Connectivity Map (CMAP), the authors proposed that targeting these pathways could sensitize cancer cells to specific drugs. This hypothesis was supported by their previous findings and corroborated with evidence from existing literature.

While the concept of the "Achilles' heels" is novel and intriguing, the manuscript fails to provide a comprehensive explanation of the synergistic mechanisms underlying the interactions between pathway inhibition and drug efficacy. This gap raises concerns about the robustness of their conclusions. Furthermore, additional issues outlined below lead me to recommend against the acceptance of this manuscript in its current form for publication in this journal.

Concerns:

1.     The authors proposed “Cancer Cell’s Seven Achilles Heels” in the manuscript title. However, their findings only supported the hypothesis that proteasome, transcription, cell cycle, and Akt-mTOR pathways are confirmed as the cell's Achilles heels.

2.     There are numerous exclusions and exceptions in the GSEA and CMAP analyses. Although the authors provided some possible reasons for these exclusions, it suggests that the analyses may not be suitable.

3.     In the Discussion section, I do not agree with the statement that "Combination therapies are viewed as most effective strategies for treating cancer patients" (Page 7, lines 206-205). Combination treatment appears to be common and effective in clinical settings, however, its success still depends on individual cases.

4.     In the Discussion section (Page 8, lines 222-231), the author argued that a milder gene downregulation was used in the study, which allowed cancer cells to survive by making them more sensitive to specific drug combinations. I am curious whether a strong shRNA downregulation of these pathways would be sufficient to induce cancer cell death. If so, there would be no need to use a combination with a more toxic drug that carries a higher risk of adverse effects.

5.     There are careless mistakes in the content, including typographical errors and incorrect phrasing.

Author Response

Michael Sherman                
Professor         
December 12, 2024 
International Journal of Molecular Sciences (IJMS) 
Dear Editor: 
Tel. +972-58-781-9472 
sherma1@ariel.ac.il 
Thank you for handling our manuscript “Cancer Cell’s Achilles Heels: Considerations for design 
of anti-cancer drug combinations” #ijms-3377637.  
We are grateful to the reviewers for their comments and corrected the problems accordingly.  
Below is a point-by-point response to the reviewers’ comments. 
In addition, we addressed the structural guidelines and reference issues that were suggested by 
both editor and reviewers. 
We hope that in the present form, the manuscript is suitable for publication. 
We look forward to hearing from you. 
Sincerely yours, 
Michael Sherman, Ph.D. 
Response to Reviewers. 
REVIEWER 2:  
Comment 1: The authors proposed “Cancer Cell’s Seven Achilles Heels” in the manuscript title. 
However, their findings only supported the hypothesis that proteasome, transcription, cell cycle, 
and Akt-mTOR pathways are confirmed as the cell's Achilles heels. 
Answer: We appreciate your feedback and have updated the manuscript title to 'Cancer Cells’ 
Achilles’ Heels' to clarify any ambiguity regarding the number of validated 'weak points' pathways.  
Comment 2: There are numerous exclusions and exceptions in the GSEA and CMAP analyses. 
Although the authors provided some possible reasons for these exclusions, it suggests that the 
analyses may not be suitable. 
Answer: We are not clear about what exclusions of exception the reviewer means. We guess that 
the question was about genes that belong to more than one pathway according to the GSEA 
software package designation. In such cases, we simply considered that they belong to the major 
core pathway found in multiple screens. We do not see how this streamlining may jeopardize the 
analysis. 
Comment 3: In the Discussion section, I do not agree with the statement that "Combination 
therapies are viewed as the most effective strategies for treating cancer patients" (Page 7, lines 
206-205). Combination treatment appears to be common and effective in clinical settings, 
however, its success still depends on individual cases. 
Answer: We agree with the reviewer that the success of the combination therapy depends on 
individual cases. However, they are heavily used in clinics and the search for new therapies is a 
major direction in the cancer field. We have addressed the issue in the discussion part (page 7, 
highlighted lines 201-202).  
Comment 4: In the Discussion section (Page 8, lines 222-231), the author argued that a milder 
gene downregulation was used in the study, which allowed cancer cells to survive by making them 
more sensitive to specific drug combinations. I am curious whether a strong shRNA 
downregulation of these pathways would be sufficient to induce cancer cell death. If so, there 
would be no need to use a combination with a more toxic drug that carries a higher risk of adverse 
effects. 
Answer: Indeed, as noted in the manuscript five of the pathways were previously reported to be 
“cancer essentiality pathways”, i.e. pathways knocking down which kills many types of cancer 
cells. However, it is clear that these pathways are essential for both cancer and normal cells (e.g. 
ribosome or proteasome pathways). The key is the degree of inhibition. In other words, cancer 
cells are more sensitive to their suppression, i.e. die at less extensive degree of knockdown. That 
is why the proteasome inhibitors are used in clinics at concentrations that inhibit protein 
degradation by 25-35%, and not 100%, since the latter will be extremely toxic to normal tissues. 
With an even lower degree of inhibition, cancer cells do not die upon inhibition of these pathways, 
but become more sensitive to other drugs. The suggestion of using monotherapies of the inhibitors 
of the weak point pathways may not be practical for multiple reasons, including potential side 
effects, pharmacological properties of the drugs, etc. 
Comment 5: There are careless mistakes in the content, including typographical errors and 
incorrect phrasing. 
Answer: Following a thorough revision of the original manuscript, we have corrected grammar 
and proofreading mistakes and eliminated typographical errors.

Reviewer 3 Report

Comments and Suggestions for Authors

Overcoming cancer remains an eternal challenge for humanity. However, the development of new anticancer drugs is extremely difficult, and their side effects are often severe, which is why drug combination therapy is commonly employed. This paper is insightful and presents remarkable findings in the context of cocktail therapy. To further enhance the value of the manuscript, I would like to offer the following suggestions.

1. Provide a more detailed explanation of the relationship between the CRISPR system and cancer therapy in the introduction.

2. Elaborate on the earlier work mentioned in the final paragraph of the introduction. Not everyone is familiar with your previous studies.

3. In Result 2.1, the global view is not well represented. Are you referring to an analysis of globally prevalent cancer types?

4. The results section titles are generally too verbose. Particularly for 2.3—did you just throw together a random sentence?

5. The current conclusion is so weak that it might be better to remove it altogether.

Author Response

Michael Sherman                
Professor         
December 12, 2024 
International Journal of Molecular Sciences (IJMS) 
Dear Editor: 
Tel. +972-58-781-9472 
sherma1@ariel.ac.il 
Thank you for handling our manuscript “Cancer Cell’s Achilles Heels: Considerations for design 
of anti-cancer drug combinations” #ijms-3377637.  
We are grateful to the reviewers for their comments and corrected the problems accordingly.  
Below is a point-by-point response to the reviewers’ comments. 
In addition, we addressed the structural guidelines and reference issues that were suggested by 
both editor and reviewers. 
We hope that in the present form, the manuscript is suitable for publication. 
We look forward to hearing from you. 
Sincerely yours, 
Michael Sherman, Ph.D. 
Response to Reviewers. 
REVIEWER 3:  
Comment 1: Provide a more detailed explanation of the relationship between the CRISPR system 
and cancer therapy in the introduction. 
Answer: We would like to clarify that our computational approach for identifying potential “weak 
points” in cancer cells was developed using data from both published and unpublished shRNA 
screens, as detailed in Results Section 2.1. Providing an in-depth discussion on the relationship 
between the CRISPR system and cancer therapy, while valuable, would shift the focus of our 
manuscript.  
Comment 2: Elaborate on the earlier work mentioned in the final paragraph of the introduction. 
Not everyone is familiar with your previous studies. 
Answer: As noted in the Introduction, we performed shRNA screens to identify genes involved in 
responses to a novel Hsp70 inhibitor JG-98 and to an antibiotic gentamycin. Such screens are 
similar to screens described in this study. We do not see what additional information about these 
original screens will be helpful to better understand our manuscript. 
Comment 3: In Result 2.1, the global view is not well represented. Are you referring to an analysis 
of globally prevalent cancer types? 
Answer: We appreciate the reviewer’s comment. Indeed, the name of the section is ambiguous. 
To address this ambiguity, we have revised the title of Section 2.1 to better reflect its focus. 
Comment 4: The results section titles are generally too verbose. Particularly for 2.3—did you just 
throw together a random sentence? 
Answer: We changed the titles of the sections to make them more focused. 
Comment 5: The current conclusion is so weak that it might be better to remove it altogether. 
Answer: The main result in our analysis is that specific pathways in cancer cells function as “weak 
points, which disturbances increase the sensitivity of cells to a variety of drugs with different 
mechanisms of action. We believe that this finding is quite novel and not trivial. The idea that 
these results may help streamline the search for synergistic anti-cancer therapies is an additional 
novelty.

Round 2

Reviewer 1 Report

Comments and Suggestions for Authors

The authors have provided accurate responses to my comments and the revised manuscript is suitable for publication now.

Reviewer 2 Report

Comments and Suggestions for Authors

The authors successfully addressed my concerns and elaborated on them in the text. I recommend accepting the revised manuscript for publication.